# Neck Dissection for Cervical Lymph Node Metastases from Remote Primary Malignancies

**DOI:** 10.3390/medicina56070343

**Published:** 2020-07-10

**Authors:** Shogo Shinohara, Hiroyuki Harada, Masahiro Kikuchi, Shinji Takebayashi, Kiyomi Hamaguchi

**Affiliations:** 1Department of Head and Neck Surgery, Kobe City Medical Center General Hospital, Minatojima-Minamimachi 2-1-1, Chuo-ku, Kobe 650-0047, Japan; stakebayashi@kcho.jp (S.T.); kiyomi_hamaguchi@kcho.jp (K.H.); 2Department of Otolaryngology, Tazuke Kofukai, Medical Research Institute, Kitano Hospital, Ohgimachi 2-4-20, Kita-ku, Osaka 530-8480, Japan; h_harada@ent.kuhp.kyoto-u.ac.jp; 3Department of Otolaryngology-Head and Neck Surgery, Graduate School of Medicine, Kyoto University, Sakyo-Ku, Kyoto 606-8507, Japan; m_kikuchi@ent.kuhp.kyoto-u.ac.jp

**Keywords:** head and neck metastasis, neck dissection, remote primary, FDG/PET-CT, survival analysis

## Abstract

*Background and Objectives*: Patients with cervical lymph node metastases from remote primary tumours have poor prognoses because of the advanced stage of their cancer. Owing to recent progress in the nonsurgical management of various cancer types, options for surgical treatment to reduce tumour volume are increasing, and may help improve survival rates. For example, neck dissection may be a good option as a definitive therapy for some patients with resectable cervical metastases. We assessed patients who underwent neck dissection with curative intent and discuss the effectiveness of this approach for cervical metastases from remote malignancies. *Material and Methods:* We retrospectively reviewed the data of 18 patients (10 males and 8 females in an age range of 30–79 years) who underwent neck dissections for neck lymph node metastases from a remote primary tumour between 2010 and 2019. Patient clinical characteristics, preoperative accuracy of positive node localisation using fluorodeoxyglucose positron emission tomography–computed tomography (FDG/PET-CT), and patient survival rates were estimated. *Results:* Primary sites included ten lungs, two mammary glands, one thymus, one thoracic oesophagus, one stomach, one uterine cervix, one ovary, and one testis per patient. There were 19 levels with FDG/PET-CT positive nodes in 17 out of 18 patients. Conversely, there were 28 pathological positive levels out of 50 dissected levels. The sensitivity, specificity, positive and negative predictive values, and accuracy of FDG-PET/CT in predicting positive nodes were 69%, 88%, 95%, 47%, and 74%, respectively. The three-year overall survival (OS) rate for all patients was 70%. The three-year OS rate of the group with zero or one pathological positive nodes was 81%, which was significantly higher than that of the group with more than two positive nodes (51%) (*p* = 0.03). *Conclusions:* Neck dissection for cervical lymph node metastases from remote primary malignancies may improve prognoses, especially considering anticancer agents and radiotherapy advancements.

## 1. Introduction

In general, cervical lymph node metastases from remote primary tumours have poor prognoses because of the advanced stage of the patient’s cancer. Head and neck surgeons will be requested to perform a cytological/pathological examination to assess cervical lymph node swelling for staging of the remote primary tumours. Fine needle aspiration cytology is generally selected at first because this procedure is minimally invasive and convenient. Sometimes, cytological examination is not enough to assess the primary organ, and a core needle biopsy/open biopsy will be required for immnohistochemical staining or genetic testing. These invasive interventions are for staging, and not with curative intent. However, advances in anticancer agents, noncytotoxic agents, and molecular targeted agents have improved the prognoses of patients with cancer with distant metastases [1,2]. In some cases, neck metastases management can improve a patient’s survival [3,4]. Hence, if other lesions, including those at the primary site, are well controlled, neck dissection may be a good option as a definitive therapy for patients with resectable cervical metastases. Furthermore, the use of whole-body fluorodeoxyglucose positron emission tomography–computed tomography (FDG-PET/CT) can help detect subclinical metastases and eradicate the rest of the tumour in any remote site of the body. However, the degree of effectiveness of neck dissection for cervical metastases from remote malignancies is still controversial [5]. In this manuscript, we retrospectively review our patients, encountered over a 10 year period, who received neck dissection for curative intent, and discuss the effectiveness of neck dissection for cervical metastases from remote malignancies.

## 2. Materials and Methods

### 2.1. Patient Data

Between January 2010 and December 2019, we performed 18 neck dissections to control neck lymph node metastases. Data on these 18 patients were retrospectively investigated. The patients comprised ten males and eight females, within an age range of 30–79 years (median age: 61 years) (Table 1). The primary sites were ten lungs, two mammary glands, one thymus, one thoracic oesophagus, one stomach, one uterine cervix, one ovary, and one testis. In 17 out of 18 cases, treatment of the primary site was completed and chemotherapy and/or radiotherapy was continued before performing the neck dissection. In one patient who underwent mammary gland dissection (case 11), primary site recurrence was simultaneously removed with axillary and cervical metastatic lymph nodes after several cycles of induction chemotherapy. In the other patient who underwent mammary gland metastases (case 12), neck dissection was simultaneously performed with axillary dissection on the same side. The laterality of the neck dissection was 10 on the left and 8 on the right. The histology of the cases, the previous treatments, and the adjuvant therapies after neck dissection are shown in Table 1. The ethics committee of Kobe City Medical Center General Hospital approved the study, and informed consent was waived because of the retrospective nature of the study.

### 2.2. Preoperative Workup

In 17 out of 18 cases, cervical lymph node metastases were primarily detected by fluorodeoxyglucose positron emission tomography–computed tomography (FDG-PET/CT) as hot spots in the neck. In one patient with mammary carcinoma, only an enhanced CT was utilised (case 12). Preoperative pathological/cytological confirmation of metastasis was only performed in five cases. Fine needle aspiration cytology was performed in three cases (case 4, 14, and 15) and positive results were obtained in two cases (cases 4 and 15). A core needle biopsy was performed for a patient with lung adenocarcinoma (case 3) because of the necessity of genetic testing at the request of a medical oncologist. An open biopsy was performed for one with a mixed germ cell tumour of the testis (case 18) after a failure of fine needle aspiration sampling. Both of them informed us of proper histological diagnoses.

### 2.3. Decision Making of the Dissection Levels in Each Case

Referring to the images of FDG-PET/CT and the clinical characteristic of the patients, the dissection levels for each case were decided in weekly tumour board meetings held by the head and neck oncology department. Ipsilateral level two to four selective dissection was selected when FDG-positive nodes were located in the jugular chain. In cases with a supraclavicular lesion in a solitary node, ipsilateral dissection at level four and five tended to be selected. In this study, the neck dissection classification proposed by the American Head and Neck society and the American Academy of Otolaryngology-Head and Neck Surgery was utilised to classify the location of lymph nodes [6].

### 2.4. Preoperative Detection Rate of Positive Nodes

To estimate the optimal neck dissection level, we compared the pathological positive nodes and radiological positive nodes detected by FDG-PET/CT preoperatively. FDG-PET/CT images were evaluated by two board-certificated radiologists/nuclear medicine physicians and one head and neck surgeon. Subsequently, focal FDG uptakes corresponding to lymph nodes identified on the CT were counted as PET-positive nodes (PET+Ns). In this study, patients without FDG-PET/CT (case 12) images were omitted for evaluation. In total, 272 nodes resected at 50 cervical lymph node levels were pathologically evaluated. Lymph nodes with nests of malignancies were defined as pathological positive nodes (*p*+Ns). The diagnostic accuracy of PET/CT in the pathological positive node assessment was estimated by comparing PET+Ns and *p*+Ns. The sensitivity, specificity, positive and negative predictive values (PPV and NPV, respectively), and accuracy were calculated according to the dissected levels.

### 2.5. Statistical Analysis

Overall survival analyses were performed using the Kaplan–Meier method. The other subanalyses were performed according to the primary tumour site (a lung cancer group and the others), pathology (an adenocarcinoma group, a squamous cell carcinoma group and the others), and the number of pathological positive nodes (divided by the median number of positive nodes). The difference in survival was compared using a log-rank test. EZR on R commander version 1.42 was used for statistical analyses, and a *p* value <0.05 was considered significant.

## 3. Results

### 3.1. Distribution of p+Ns according to Primary Site

The distribution of the dissected level and the level with the *p*+N values is shown in Table 2. The most dissected level was level four (inferior jugular chain) on the left side, followed by level three (middle jugular chain) on the left and level four on the right. The most pathological positive level was also level four on the left side, followed by level four on the right side. Metastases from the lung, mammary gland, and upper GI occurred in both sides, whereas lesions on organs within the pelvic (ovary, cervix and testis) region tended to occur only on the left side.

### 3.2. Diagnostic Accuracy of PET+Ns for Predicting p+Ns

The distribution of PET+Ns and *p*+Ns for each of the patients is shown in Table 3. There were 24 PET+Ns in 19 levels in 17 out of 18 patients (FDG-PET/CT was not performed in case 12). However, there were 80 *p*+Ns (29%) out of 272 dissected nodes in 28 (56%) out of 50 levels. One patient with a mixed germ cell tumour that originated from the testis (case 18) showed no viable tumour cells in the dissected nodes, despite the presence of an FDG-hot spot at level four because of additional chemotherapy after FDG/PET-CT and open biopsy. The distribution of *p*+N levels and PET+N levels according to the neck level is shown in Table 2. In this table, numerators indicate the number of PET+N cases, whereas denominators indicate the number of *p*+N cases at each level. The sensitivity, specificity, PPV, NPV and accuracy of FDG-PET/CT for predicting *p*+N were 69%, 88%, 95%, 47%, and 74% evaluated per level (Table 4).

### 3.3. Survival

The median time to death or last follow-up of 18 patients was 41 months. Recurrence occurred in nine patients, six of whom experienced lung/mediastinum recurrence, four experienced neck recurrence, two abdominal recurrence, and one bone recurrence. Two patients were managed successfully with radiotherapy and/or additional chemotherapy. Seven deaths occurred because of cancer. Among the four cases that recurred in the neck, two (case 11 and 12, cases with mammary gland cancer) experienced recurrence at the lower border of the supraclavicular nodes on the dissected side, just behind the clavicle. The rest of the patients had recurrence on the contralateral sides (level 3 in case 14 and level 4 in case 16) of the neck.

The three-year overall survival (OS) rate for all patients was 70%. Patients with lung cancer had a better OS rate (77%) than those with the other types of malignancies (63%); however, the difference was not significant (*p* = 0.24) (Figure 1). The three-year OS rates for each pathological group were 74% in the adenocarcinoma group, 40% in the squamous cell carcinoma group and 100% in the other histological group (Figure 2). The number of pathological positive nodes (*p*+N) ranged from 0–20 (the median number was 1), and when all patients were dichotomised, seven had more than 2 *p*+N and 11 had 0 or 1 (Table 3). The three-year OS rate of the group with less than 2 *p*+N was 81%, which was significantly higher than that for the group with more than 2 *p*+N (51%) (*p* = 0.03) (Figure 3).

## 4. Discussion

Neck dissection is an essential surgical procedure for the treatment of malignant tumour metastases originating from primary head and neck lesions. Sometimes, prophylactic neck dissection is considered for locally advanced head and neck malignancies because they are often accompanied by subclinical metastases. Compared with the “wait and see” strategy, neck dissection improves patient survival [7,8]. However, the effectiveness of neck dissection for cervical metastases from remote malignancies is still controversial. For some malignancies, cervical lymph nodes are categorised as regional. In those with mammary gland carcinoma, ipsilateral supraclavicular lymph nodes are categorised as regional, whereas patients with supraclavicular lymph node metastases are classified as N3 and with at least stage IIIC cancer. In those with nonsmall cell lung carcinoma, patients with ipsilateral supraclavicular metastases are classified into N3 and staged IIIB or C. For those with oesophageal carcinoma, the staging system varies according to national guidelines of each country. When lymph nodes in the upper jugular chain are affected, the lesions are categorised into distant metastases according to the TNM/AJCC oesophageal cancer classification system. According to the Japanese classification of oesophageal cancer, lymph nodes in the upper jugular chain are numbered as 102 and categorised into N3 regional lymph nodes in cases where the primary site is the upper thoracic region [9]. In our case (case 14), the primary site was in the upper thoracic cavity, and the detected node was categorised into 102 in the guidelines for Japanese surgeons.

In our study, patients with lung cancer metastases, especially those with adenocarcinoma of the lung, were mostly treated by undergoing surgical neck dissection. This may be because of recent progress in anticancer agents in the field of lung cancer [10]. Investigations into the genes responsible for lung cancer are helping to identify the most suitable anticancer agents based on a patient’s genetic makeup [11,12]. Thanks to this progress, some patients showed a residue of the tumor only in the cervical area after several lines of chemotherapy and had a chance of survival by removing it surgically. However, there have been no reports on recommendations for neck dissection for cervical metastases from a lung carcinoma as a routine surgical strategy. As some surgeons perform neck dissections with curative intent for patients with isolated supraclavicular lymph node during concurrent chemotherapy [13], more reports are expected in the future. In our assessment of patients with lung cancer, the three-year OS was relatively good (76%), although all the patients were categorised into more than stage IIIB.

In patients with mammary cancer, neck metastases occurrence has been reported to be 2–4% [14,15,16]. Cancer cells from mammary glands travel in the lymphatic duct along the axillary and subclavian veins into the jugulosubclavian angle, and, in this process, cervical lymph node metastases occur in the supraclavicular lesion. There are currently only a few previous studies that refer to the impact of neck dissection on patient survival [16,17]. Brito et al. [17] conducted a study which proved that 70 patients with supraclavicular lymph nodes who received induction chemotherapy, surgery, and postoperative chemoradiotherapy showed significantly better survival than that among other patients with a stage IV disease. After this report, patients with supraclavicular lymph node involvement were downgraded to stage IIIB. Bisase et al. concluded from a survey of 117 head and neck surgeons in United Kingdom that there is wide-spread inconsistency in patients’ management with cervical metastases of mammary cancer, but there was a trend towards aggressive surgical treatment despite the lack of high-level evidence [18]. In a recent retrospective study based on 78 Korean patients with ipsilateral supraclavicular lymph node metastases of mammary cancer, it was revealed that neck dissection did not improve loco-regional or disease free survival [19]. In the current Japanese clinical treatment guidelines for patients with breast cancer, surgical treatment of supraclavicular metastatic lesion was not recommended (weak evidence level) [20]. In our series, both of the patients with this presentation died from the disease, two and four years following treatment. We also had recurrences in the area behind the clavicle in both cases, which were not detected in the preoperative imaging. We had expected postoperative radiotherapy to exterminate the microscopic residue of cancer cells in the blind area between axillary and supraclavicular dissections, but this was not the case.

Among the remote primary lesions of organs in the pelvic region which cause cervical metastases, testicular cancer is known to have an indication of neck dissection after induction chemotherapy. The incidence of cervical metastasis of testicular cancer was reported to be 4.5% with a neck mass described as the first symptom in 5% of cases [21]. Gupta et al. reported on their 968 patients with testicular carcinoma and identified 41 patients who underwent postchemotherapy neck resection due to residual lesions [3]. They concluded that residual neck mass resection leads to excellent local control and can contribute to long-term disease control and survival. In this study, we experienced one patient with testicular carcinoma who received neck dissection after systemic chemotherapy and who had been alive with no evidence of disease for over eight years. For those who had ovarian or cervical carcinoma, there currently is no evidence to support neck dissection [5].

In the field of oesophageal carcinoma, there are a wide variety of treatment strategies that differ based on national recommendations per country. Kato et al. found that patients who underwent bilateral neck dissection, besides mediastinum and abdominal lymphadenectomy during the transthoracic esophagectomy, had a 15% survival benefit at five years following treatment compared to those who did not undergo bilateral neck dissection [4]. Following publication of this report, additional neck dissection in the treatment of patients with thoracic oesophageal carcinoma (esophagectomy with three-field lymph node dissection) has become a standard strategy in some countries [22,23]. For patients with gastric carcinoma, metastasis to the left sided nodes in the neck is known as Virchow’s node, named after the famous pathologist, Rudolf Virchow. Unfortunately, the presence of Virchow’s node indicates poorer disease prognoses, as survival rates are approximately 4% at five years following treatment [5]. In this study, we had one patient with gastric carcinoma who received neck dissection after several cycles of chemotherapy. He received adjuvant chemotherapy after neck dissection and has been alive with no evidence of disease for over seven years.

The emergence of FDG-PET/CT has been associated with great progress in the staging process of many types of malignancies. FDG-PET/CT is expected to detect distant metastatic lesions better than conventional modalities because it can assess the metabolic state and volume of the tumour. For those with head and neck cancer, the accuracy of FDG-PET/CT for detecting cervical metastases was reported to be 92–93% on a level-by-level basis, i.e., significantly higher than that of CT/MRI [24]. However, the limitations of node staging using FDG-PET/CT have also been well documented, especially for assessing those with oral cancer. Because of the poor prognoses of patients with cervical metastases after treatment, prophylactic neck dissection was justified in some patients with oral cancer who had FDG-PET/CT negative neck node lesion assessments [7,8]. The finest spatial resolution of a PET-CT scanner (4–6 mm) limits its sensitivity for microscopic disease that is detectable only by histopathological examination after neck dissection [25,26]. For our study, FDG-PET/CT was utilised for cervical metastases detection in most cases. The specificity of the FDG-positive nodes was high (88%), while the sensitivity and accuracy were not satisfactory (69% and 74%, respectively). This was mainly because of the variation of histological types/primary sites in this series of cases. The intensity of FDG uptake in the metastatic cervical lymph nodes differed significantly among cases. We had to largely depend on the examiners’ experience to determine PET positive/negative, which was a limitation of this study. Although we performed neck dissections that included additional levels in addition to the detected levels, 4 out of 18 patients developed neck recurrence. Preoperative assessments using other modalities, such as ultrasonography, might be a good option to determine optimal levels for surgical resection [27,28]. Moreover, only the number of pathological positive nodes was a potential prognostic factor in this study. Cases with solitary positive nodes (or pathological negative nodes) had significantly better OS than those with multiple pathological positive nodes. The accuracy of the preoperative assessment for the number of positive nodes may be more important in determining whether neck dissection is appropriate for metastases that originated from remote malignancies.

## 5. Conclusions

As progress is made in the nonsurgical management of patients with cancers, such as the availability of noncytotoxic chemotherapy, molecular targeted agents, and immune checkpoint inhibitors, the indication of surgical treatment to reduce tumour volume will likely expand and help to improve patient survival. Neck dissection for cervical lymph node metastases from remote primary malignancies will be performed more in the future. However, evidence-based recommendations for optimal management are lacking because cervical metastases from remote primary malignancies are quite rare. Patients with lymph node metastases to the neck are generally at an advanced stage of cancer, and the goals of the treatment are to improve quality of life and obtain better survival benefit. Therefore, multidisciplinary cancer teams and environments, along with case-by-case discussions, will be indispensable for providing optimal care.

## Figures and Tables

**Figure 1 medicina-56-00343-f001:**
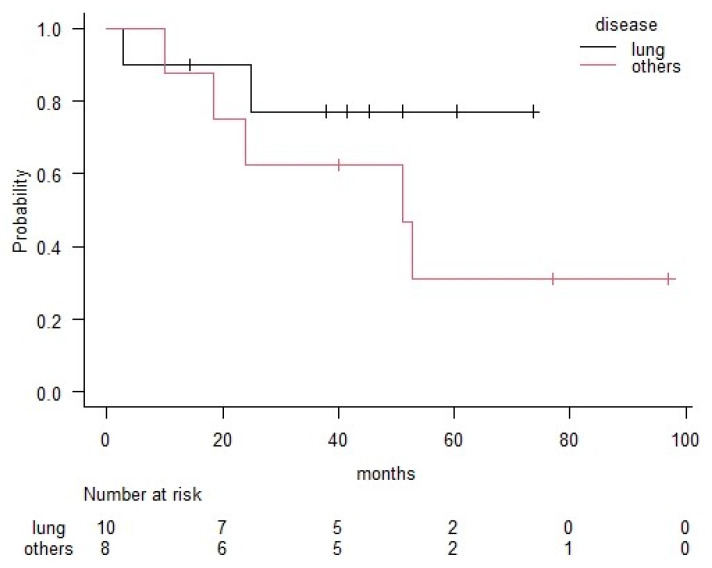
Comparison between the lung cancer group and the other group in terms of overall survival according to the Kaplan–Meier method. Significance was based on log-rank comparisons. Significant differences were not observed between these two groups (*p* = 0.24).

**Figure 2 medicina-56-00343-f002:**
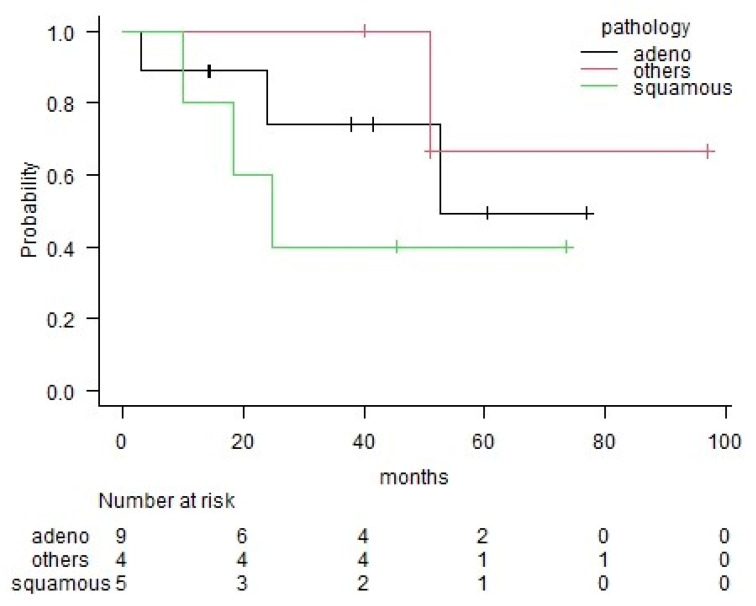
Comparison among adenocarcinoma, squamous cell carcinoma, and other pathological type groups in terms of overall survival according to the Kaplan–Meier method. Statistical analyses were not performed because of the small number of cases.

**Figure 3 medicina-56-00343-f003:**
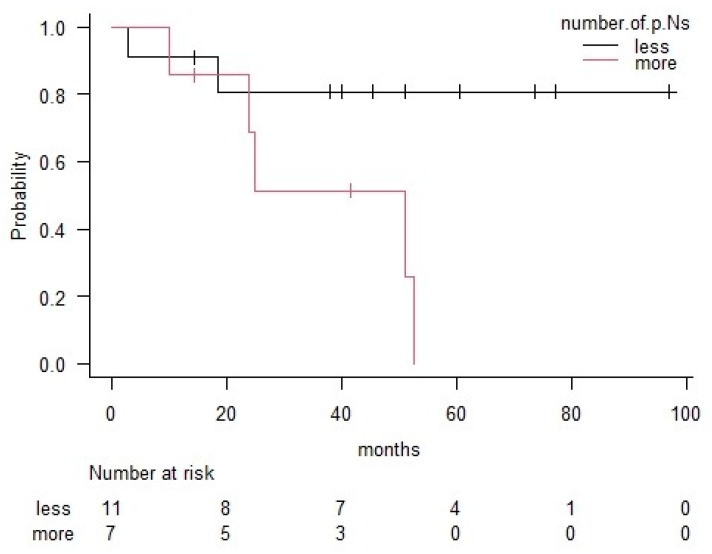
Comparison between less pathological positive node (0–1) group and more pathological positive node (no less than two) group in terms of overall survival according to the Kaplan–Meier method. Significance was based on log-rank comparisons. A group with less pathological positive nodes had significantly better survival than the group with more pathological positive nodes (*p* = 0.03). p.Ns: pathological positive node.

**Table 1 medicina-56-00343-t001:** Patients’ characteristics.

Cases	Age	Sex	Primary	Histology	Previous Treatment			Dissected Side	Adjuvant Therapy	
					Surgery	CT	RT		CT	RT
1	49	M	Lung	Adeno.		◯	◯	L	◯	
2	61	M	Lung	Adeno.		◯	◯	L		
3	67	M	Lung	Adeno.		◯		R	◯	
4	71	M	Lung	Adeno.		◯	◯	L	◯	
5	41	F	Lung	Adeno		◯	◯	L	◯	
6	49	F	Lung	Adeno	◯	◯	◯	R	◯	
7	75	F	Lung	Adeno-squamous		◯	◯	R		
8	51	M	Lung	SCC		◯	◯	L	◯	◯
9	63	F	Lung	SCC	◯	◯	◯	R	◯	◯
10	63	M	Lung	SCC	◯	◯	◯	R		
11	47	F	Mammary	Adeno.	◯	◯	◯	R	◯	◯
12	73	F	Mammary	Adeno.	◯	◯	◯	L	◯	◯
13	55	M	Thymus	Malignant thymoma	◯	◯	◯	R		
14	60	M	Esophagus	SCC	◯	◯		R		◯
15	69	M	Stomach	Adeno.	◯	◯		L	◯	
16	79	F	Cervix	SCC		◯	◯	L	◯	
17	42	F	Ovary	Endometrioid adeno.	◯	◯		L	◯	
18	30	M	Testis	Germ cell tumor	◯	◯		L		

Adeno: adenocarcinoma; SCC: squamous cell carcinoma; CT: chemotherapy; RT: radiotherapy; L: left; R: right; ◯: yes; blank: no.

**Table 2 medicina-56-00343-t002:** Distribution of PET+Ns and *p*+Ns according to the level of the neck.

	Lung	Mammary	Thymus	Upper GI	Pelvic	Total
L2	1/1			0/1		1/2
L3	1/5	1/1		0/1	1/2	3/9
L4	4/5	1/1		1/1	2/3	8/10
L5	1/3	1/1			1/2	3/6
R2	0/1			1/1		1/2
R3	1/4	1/1	0/1	0/1		2/7
R4	5/5	1/1	1/1	0/1		7/8
R5	2/4	1/1	0/1			3/6
Total	15/28	6/6	1/3	2/6	4/7	28/50

L: left; R: right. Numerators indicate the number of PET+Ns cases and denominators indicate the number of *p*+Ns cases in each level. Upper GIs (upper gastrointestinal tracts) consists of a case with thoracic oesophagus and one with stomach. Pelvic lesions (Pelvic) consist of a case with ovary, cervix, and testis. PET+Ns: PET-positive nodes; *p*+Ns: pathological positive nodes; PET: positron emission tomography.

**Table 3 medicina-56-00343-t003:** Distribution of PET+Ns and *p*+Ns according to the patients.

Case	Age	Sex	Primary	Histology	Number of LNs		Number of Levels	
					PET+Ns	p+Ns	PET+Ns	p+Ns
1	49	M	Lung	Adeno.	1	1	1	1
2	61	M	Lung	Adeno.	1	1	1	1
3	67	M	Lung	Adeno.	2	1	1	1
4	71	M	Lung	Adeno.	2	7	2	3
5	41	F	Lung	Adeno.	1	1	1	1
6	49	F	Lung	Adeno.	1	10	1	2
7	75	F	Lung	Adeno-squamous	1	1	1	1
8	51	M	Lung	SCC	1	1	1	1
9	63	F	Lung	SCC	3	8	1	3
10	63	M	Lung	SCC	1	1	1	1
11	47	F	Mammary	Adeno.	2	3	2	3
12	73	F	Mammary	Adeno.	N/A	17	N/A	3
13	55	M	Thymus	Malignant thymoma	3	5	1	1
14	60	M	Esophagus	SCC	1	1	1	1
15	69	M	Stomach	Adeno.	1	1	1	1
16	79	F	Cervix	SCC	1	20	1	3
17	42	F	Ovary	Endometrioid adeno.	1	1	1	1
18	30	M	Testis	Germ cell tumor	1	0	1	0
				Total	24	80	19	28

Adeno: adenocarcinoma; SCC: Squamous cell carcinoma; LNs: lymph nodes: N/A: Not available; PET+Ns: PET-positive nodes; *p*+Ns: pathological positive nodes.

**Table 4 medicina-56-00343-t004:** Comparison of PET+Ns and *p*+Ns per level.

		Pathology		
		Positive	Negative	Total
PET	positive	18	1	26
	negative	8	7	8
	total	19	15	34

Sensitivity = 69%, Specificity = 88%, Positive predictive value (PPV) = 95%, Negative predictive value (NPV) = 47%, Accuracy = 74%.

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
