# Peer review of "Neck Dissection for Cervical Lymph Node Metastases from Remote Primary Malignancies"

_medicina, 2020, doi:10.3390/medicina56070343_

Round 1

Reviewer 1 Report

Interesting article, about an always difficult situation. The authors make a retrospective analysis of a relative large cohort of patients with distant metastasis in the neck.

There are two points that I consider need to be clarified. 

  1. When you talk about LN biopsy in your introduction, please be more precise. Did you mean Fine needle aspiration biopsy or trucut? open biopsy? this need to be clarified because this journal have a wide range of reader, not just H&N surgeons. 
  2. The authors needs to clarify why the conduct the workup in this way. In the case of open byopsy (Usually neglected for H&N metastasis) was after a failed tru-cut of FNAB?

Thanks.

Author Response

Response to reviewer’s comment 1

  1. When we talk about LN biopsy in our introduction, we should be more precise.

Did we mean fine needle aspiration biopsy, trucut or open biopsy when we used a word “biopsy”?

This needs to be clarified because this journal has a wide range of reader, not just H&N surgeons.

(Response)

Thank you for your thoughtful comment to our manuscript.

I changed the description in introduction as follows.

“Head and neck surgeons will be requested to perform a cytological/pathological examination to assess cervical lymph node swelling for staging of the remote primary tumours. Fine needle aspiration cytology is generally selected at first because this procedure is minimally invasive and convenient. Sometimes cytological examination is not enough to assess the primary organ and core needle biopsy or open biopsy will be required for immnohistochemical staining or genetic testing. Any of these invasive interventions are for staging but not for curative intent.”

  1. We need to clarify why we conduct the workup in this way. In the case of open biopsy (usually neglected for H&N metastasis), was it done after a failed tru-cut of FNAB?

(Response)

Thank you for your comment.

We explained in detail about the cases with core needle biopsy and open biopsy.

We changed the description in this part as follows.

“A core needle biopsy was performed for a patient with lung adenocarcinoma (case 3) because of the necessity of genetic testing by request of medical oncologist. Open biopsy was performed for one with a mixed germ cell tumour of the testis (case 18) after a failure of the fine needle aspiration sampling. Both of them informed us proper histological diagnoses.”

Reviewer 2 Report

This manuscript presents to investigate the effectiveness of surgical treatment for cervical metastases from remote malignancies. The cohort consists of malignancies originating from different organs, which makes it difficult to draw substantial conclusions. Not surprisingly, biological behaviors of tumors are different from disease to disease. The several facts described in this article may certainly be of use for other head and neck surgeons.

L.95 “FDG-PET/CT images were evaluated by two board certificated radiologists/nuclear medicine physicians and one head and neck surgeon.”

Uptakes of FDG can be affected by various factors, such as tumor sizes, blood glucose levels, imaging devices, and so on. To be sure, it is difficult to determine the uniform valuation basis, and that is why FDG-PET/CT images should be evaluated by experts of nuclear medicine in clinical practice. However, with respect to the scientific investigation about the performance of this method for preoperative detection of malignancies, I think the authors need to clearly articulate some kind of criterion of “FDG-PET positive”, especially if the authors consider that the accuracy of the preoperative assessment of the number of positive nodes is important for deciding an indication of neck dissection originated from remote malignancies.

L.150 “Two experienced recurrence at the lower border of the supraclavicular nodes on the dissected side, just behind the clavicle.”

Did you detect these lymph nodes in the preoperative images? If there is anything that should be improved with regard to the area of the neck dissection, or to the indication of surgery, it would be beneficial to state about the opinions of the authors.

L.35 “Neck dissection for cervical lymph node metastasis from remote primary malignancies may improve prognosis of patients, especially considering anticancer agent and radiotherapy advancements.”

L.197 “This may because of recent progress in anticancer agents in the field of lung cancer. Investigations into genes responsible for lung cancer are helping to identify the most suitable anticancer agents based on a patient’s genetic makeup, and thanks to this progress, some patients maintain complete response when primary and thoracic lymph nodes are involved.”

These discussions are difficult to follow. If therapeutic agents for lung cancer achieved a remarkable development, I think this will reduce the necessity of surgical neck dissection.

Author Response

Response to reviewer’s comment 2

  1. Uptakes of FDG can be affected by various factors, such as tumor sizes, blood glucose levels, imaging devices, and so on. To be sure, it is difficult to determine the uniform valuation basis, and that is why FDG-PET/CT images should be evaluated by experts of nuclear medicine in clinical practice. However, with respect to the scientific investigation about the performance of this method for preoperative detection of malignancies, I think the authors need to clearly articulate some kind of criterion of “FDG-PET positive”, especially if the authors consider that the accuracy of the preoperative assessment of the number of positive nodes is important for deciding an indication of neck dissection originated from remote malignancies.

(Response)

Thank you for your thoughtful comment to our manuscript.

I agree with your opinion but it was so hard to articulate criterion of FDG-PET positive in this study. There were different kinds of histologic types/primary sites and the intensity of FDG uptakes differed among them, for example, a case with metastatic adenocarcinoma from lung indicated fainted uptakes (in case 6) while a case with metastatic endometrioid adenocarcinoma showed strong uptakes (in case 17). We cannot help relying on the judge of the experts to determine FDG positive/negative.

I added this issue in the discussion part as a limitation of this study.

“This was mainly because of the variation of histological types/ primary sites in this series of the cases. The intensity of FDG uptakes in the metastatic cervical lymph nodes differed a lot among cases. We had to largely depend on the examiners experience to determine PET positive/negative, and this was a limitation of this study.”

2) Did you detect these lymph nodes in the preoperative images? If there is anything that should be improved with regard to the area of the neck dissection, or to the indication of surgery, it would be beneficial to state about the opinions of the authors.

(Response)

Thank you for your educational comment to our manuscript.

We did not detect those lymph nodes in the preoperative imaging. However, we supposed microscopic residue of cancer cells in these cases because axillary dissections performed simultaneously and the area behind the clavicle was a blind area of the spread of the cancer from axillary area to the suplaclavicular area. We had done postoperative radiotherapy in both of the cases but we got recurrences.

We added the following sentences for explaining about this issue.

“We also had recurrences in the area behind the clavicle in both cases, which were not detected in the preoperative imaging. We had expected postoperative radiotherapy to exterminate the microscopic residue of cancer cells in this blind area between axillary and supraclavicular dissections but failed.”

3) L.35 “Neck dissection for cervical lymph node metastasis from remote primary malignancies may improve prognosis of patients, especially considering anticancer agent and radiotherapy advancements.”

L.197 “This may be because of recent progress in anticancer agents in the field of lung cancer. Investigations into genes responsible for lung cancer are helping to identify the most suitable anticancer agents based on a patient’s genetic makeup, and thanks to this progress, some patients maintain complete response when primary and thoracic lymph nodes are involved.”

These discussions are difficult to follow. If therapeutic agents for lung cancer achieved a remarkable development, I think this will reduce the necessity of surgical neck dissection.

(Response)

Thank you for your comment to our manuscript.

I really appreciate your comment discussing seriously about this issue.

The large part of our cases with lung cancer had received multiple lines of chemotherapeutic agents before the doctors asked us to remove only the residue of the systemic metastases in the neck. Without the progress of the chemotherapeutic agents, they would not have a chance to survive and they would not be introduced to us, head and neck surgeons.

Please allow us to explain about it

We changed and added following description in the discussion part.

“and thanks to this progress, some patients show a residue of the tumor only in the cervical area after several lines of chemotherapy and have a chance to survive by removing it surgically.”
